# COVID-19 Vaccination and Vaccine Hesitancy in the Gaza Strip from a Cross-Sectional Survey in 2023: Prevalence, Risk Factors, and Associations with Health System Interventions

**DOI:** 10.3390/vaccines12101098

**Published:** 2024-09-26

**Authors:** Jennifer Majer, Jehad H. Elhissi, Nabil Mousa, Jill John-Kall, Natalya Kostandova

**Affiliations:** 1International Medical Corps, Los Angeles, CA 90025, USA; jjohnkall@internationalmedicalcorps.org (J.J.-K.); nkostandova@internationalmedicalcorps.org (N.K.); 2Department of Medicine, Al-Azhar University, Gaza City P850, Palestine; jehadelhissi@gmail.com; 3Programs Department, International Medical Corps, Gaza City P850, Palestine; nabilmousa@internationalmedicalcorps.org

**Keywords:** COVID-19, vaccine, hesitancy, Gaza Strip, Palestine, healthcare workers

## Abstract

Background: Preventing COVID-19 in Gaza is crucial due to the devastation of advanced health services infrastructure by war. Despite the high protection offered by COVID-19 vaccines against severe disease, a 2021 survey in Gaza found only half of the population was vaccinated, and one-third was vaccine-hesitant. This follow-up study conducted in March 2023 aimed to re-evaluate vaccination levels, hesitancy, exposure to vaccine promotion efforts, and other risk factors in Gaza. Methods: A community-based cross-sectional survey with multistage stratified sampling was used. Associations of primary exposures and other determinants with vaccine status and hesitancy were quantified using bivariate and multivariable logistic regression. Results: In 2023, 63.5% of adults received at least one vaccine dose compared to 49.1% in 2021 (*p* < 0.001). Vaccine hesitancy prevalence was 31.7% in 2023 versus 34.1% in 2021 (*p* = 0.395). Adjusted odds of vaccination were 4.2 times higher among those referred by health workers compared to those not referred. Adjusted odds of vaccine hesitancy among those who received information on the vaccine from health workers were 0.3 times that of people who did not receive information. Conclusions: Results suggest health workers could play a crucial role in future vaccination strategies, as their vaccine promotion efforts were linked to better vaccine outcomes. Investing in the skills development of community health workers to contribute to these efforts is recommended.

## 1. Introduction

The October 2023 war in the Gaza Strip has had a catastrophic impact on the population’s health and healthcare system. Scenario-based projections from February 2024 estimated COVID-19, influenza, and pneumococcal diseases would be the leading causes of excess mortality in Gaza from endemic infectious diseases [1]. In Gaza, where only one-quarter of the territory’s primary healthcare facilities and one-third of hospitals were reported to be functional in April 2024 [2], prevention is vital due to the limited access to advanced healthcare services for patients in respiratory distress. COVID-19 vaccines are highly effective in preventing severe disease and mortality due to SARS-CoV-2 virus [3,4], and have an important role in reducing excess deaths in Gaza, especially as access to healthcare services continues to decrease. However, this strategy is heavily dependent on achieving high vaccination rates in the population and in vulnerable sub-groups.

Data collected prior to the war can provide an evidence base for vaccine interventions. Two recent population-based surveys in Gaza in 2021 and 2023 assessed the COVID-19 vaccine coverage, vaccine hesitancy, and associated risk factors. The first survey was carried out in October 2021, at the beginning of a COVID-19 response program implemented by International Medical Corps (IMC) alongside local healthcare partners. In 2021, the population-weighted vaccine prevalence was estimated to be 49.1% (95% CI: 43.1–55.1) in the adult population [5]. Vaccine hesitancy was 34.1% (95% CI: 28.1–40.6) in the pooled population of vaccinated and unvaccinated individuals. Disparities in vaccine uptake were observed, with women, younger individuals, and those with less education reporting lower vaccination rates and higher levels of vaccine hesitancy. Higher confidence in the vaccine’s safety and larger self-perceived health risks of COVID-19 (including the likelihood of contracting the virus and the risk of severe illness) were linked to lower levels of vaccine hesitancy and higher vaccine uptake. These trends have been observed in other settings [6,7].

The 2023 survey re-evaluated the levels of vaccination, hesitancy, and factors associated with vaccine outcomes, including both risk factors and exposure to vaccine promotion efforts through the health system. A key component of the COVID-19 response program was to promote vaccination through referrals and awareness raising by health workers. In our study, we distinguish between formal healthcare workers (HCWs) and community health workers (CHWs). HCWs are formally trained providers, such as physicians, nurses, and midwives. In contrast, CHWs are community members who, after receiving basic non-clinical training, work either voluntarily or for a stipend to support public health messaging and case identification [8,9].

Evidence generally suggests that the source of information is a significant driver of vaccine behaviors [10,11]. However, the effectiveness of vaccine promotion by HCWs and CHWs remains under-researched, particularly in low-resource settings and with vaccines targeting adult populations. In a convenience sample from the UK, US, and Turkey, higher levels of COVID-19 vaccine uptake were reported among individuals who received vaccination messages delivered by expert scientists, whose friends and family members were vaccinated, and who received risk-based incentives [12]. It remains unclear whether health workers in Gaza can influence vaccine acceptability and uptake. This gap in knowledge was a supplementary topic we explored in our follow-up survey.

To evaluate program outcomes and address these broader evidence gaps, this study had three primary objectives: (1) To provide updated prevalence estimates of COVID-19 vaccination and vaccine hesitancy among the adult population of Gaza; (2) to describe overall and strata-specific prevalence of exposure to vaccine promotion by healthcare workers and risk factors for poor vaccine outcomes; and (3) to investigate whether vaccine outcomes were associated with exposure to vaccine promotion and other hypothesized risk factors.

## 2. Materials and Methods

### 2.1. Sampling Design and Sample Size

We used a community-based cross-sectional survey with a multistage stratified sampling design to select households from all five governorates (Figure 1). Within each governorate, we selected one geo-locality from each of three strata (urban, refugee camps, and rural) using probability proportional to size, with size in each geo-locality based on 2021 population projections from the Palestinian Bureau of Central Statistics (PBCS) (Appendix A) [13]. In the second stage of sampling, 123 clusters (administrative units within the geo-localities comprising blocks in the camp settings and neighborhoods in the urban and rural areas) were randomly selected, with the number of clusters selected per geo-locality proportional to their size. In the third stage, households within a cluster were selected using systematic random sampling. However, enumerators used convenience sampling at the household level and interviewed the first point of contact of those who were available (at home), willing (consented), and eligible (adults 18 years or older) to participate.

Sampling weights were developed to correct for differences in the sample distribution due to stratification compared with the target population and the multistage sampling design (selection of geo-locality, cluster, and household). However, unlike the 2021 survey, the data were not weighted at the cluster level within the geo-localities because the cluster codes corresponding to each household were not available in the original dataset and could not be verified due to the war.

For the 2023 survey, the sample size was calculated using EpiInfo (CDC) (v7.2.4.0) [14] to detect a difference in vaccine hesitancy of 0.12 between 2021 and 2023, based on the program target of 0.24 (36% at baseline), with 80% power. The parameters for the sample size calculation were set at a 5% significance level, a non-response rate of 5%, and a design effect of 2. The calculation yielded a sample of 922 households.

### 2.2. Data Collection

The survey was conducted using tablets with the questionnaire deployed on KoboCollect [15]. Ten survey enumerators (five men and five women) were recruited, each of whom had a public health background and prior experience with data collection. They were trained by the survey consultant prior to data collection. Interviews were conducted in person between 5 and 23 March 2023. Oral informed consent was obtained from all participants prior to the interviews. All data collection in the field was supervised by a survey consultant and two staff members from the IMC’s monitoring, evaluation, accountability, and learning (MEAL) department.

### 2.3. Study Variables

Table 1 summarizes the thematic areas covered in the survey, the corresponding variables used in the analyses, and the questions from the survey. The primary outcomes were “COVID-19 vaccination status” and “COVID-19 vaccine hesitancy”. For comparison with the 2021 survey, we used the same set of questions to assess the primary vaccine outcomes and risk factors. Vaccination status was assessed through the question, “Have you already received the COVID-19 vaccine?”. The number of doses and manufacturer were also requested for people who confirmed the vaccination. Respondents who received at least one dose of any COVID-19 vaccine were considered vaccinated.

Vaccine hesitancy was assessed based on the SAGE working group definition of vaccine hesitancy using the question, “If you could get a COVID-19 vaccine this week, would you get it?”. Non-vaccinated individuals were classified as hesitant if they responded “no” or “unsure”, while all vaccinated individuals were classified as non-hesitant.

To assess the association between vaccine promotion by health workers and vaccine outcomes, two primary exposures were used: (1) referrals to take the vaccine and (2) receipt of information on the vaccine. Referral was assessed through the question, “Have IMC partners, health organizations and/or NGOs referred you to get vaccinated?” Receipt of information was assessed through the question, “Have you received information about vaccination from IMC partners, health organizations and/or NGOs?”

Risk factors included in the analyses were demographic characteristics and selected determinants of vaccine hesitancy based on our previous study. Fixed characteristics were sex, age group, and highest level of education attained. Age was collapsed into two levels (18–39 and 40+), and education was collapsed into three levels (primary or none, secondary or vocational, and university) for regression models. Modifiable risk factors were proxy variables for trust in the health system, trust in HCWs, trust in CHWs, vaccine acceptability, and perceived individual risk from COVID-19. See Table 1 for variable definitions and the survey questions used.

Out of the total sample (*n* = 919), we excluded from the analyses any respondents who did not have recorded outcomes for vaccine status (*n* = 2), hesitancy (*n* = 22), or both (*n* = 1), resulting in a final sample size of 894.

### 2.4. Statistical Analysis

Univariate descriptive statistics were used to summarize the demographic characteristics of the sample population, vaccine coverage and hesitancy outcomes, and risk factors. Estimates were presented as overall point estimates and stratified by location of residence (geographic strata). All stratified estimates and survey-specific estimates (2021 vs. 2023) were compared using chi-square test statistics, and the tables included *p*-values and 95% confidence intervals (CI).

Bivariate and multivariable logistic regressions were used to quantify the associations of the primary exposure, and a selected set of other determinants with vaccine status and vaccine hesitancy, where referral to vaccination was the primary exposure for vaccine status and receipt of information was the primary exposure for hesitancy. For comparability and consistency with the earlier study [5], the same risk factors were included in the multivariable models—age, sex, highest educational attainment, confidence in the safety of the vaccine, trust in health workers to provide accurate information on the vaccine, and perception of severe disease risk.

All statistics presented represent population-weighted estimates. For the 2023 survey, we could not account for sampling at the cluster level as cluster information was not available. Analyses were conducted using R version 4.3.2 [16] and RStudio version 2023.12.1 + 402 [17] with the packages survey [18] for all statistics, ggplot2 [19] for figures, and gtsummary [20] for the tables.

## 3. Results

### 3.1. Characteristics of Survey Respondents

The demographic characteristics of survey respondents are shown in Table 2. Younger adults, 18–39-year-olds, represented 56% of the sample, and 45% of respondents were female. Close to one-fifth of the respondents reported having primary or no formal schooling. The largest proportion of respondents was from Gaza City (34%), and the lowest was from Rafah (12.5%). The governorate-level distribution of the 2023 sample aligns with PCBS 2021 population projections for the Gaza Strip [13]. However, the sample somewhat over-represented urban areas compared to PCBS projections. The survey sample was comprised of 83% urban residents and 12.6% camp residents, whereas the PCBS for Gaza estimated 73% for urban residents, 23% for refugee camps, and 4.5% for rural areas.

### 3.2. Vaccine Outcomes

#### 3.2.1. Vaccination Status

The percentage of adults who received at least one dose of the vaccine was 63.5% (95% CI: 59.4–67.5) in 2023 compared with 49.1% (95% CI: 43.1–55.1) (*p* < 0.001) in 2021 (Figure 2) (Appendix A). Vaccine uptake was the highest among residents of refugee camps (73.9%, 95% CI: 68.1–79.7), followed by rural areas (71.2%, 95% CI: 62.3–80.2), and was lowest among urban residents (61.5%, 95% CI: 56.7–66.3) (*p* = 0.001) (Table 3).

Details of respondents’ vaccination history are provided in Appendix A. The majority of vaccines administered were Pfizer (63.6%) and Sputnik (24.7%), either alone or in combination. To be classified as fully vaccinated, at least two doses were required. A sizable proportion were only partially vaccinated, as 43.5% of respondents reported receiving one dose, 48.6% reported receiving two doses, and 7.9% reported receiving three doses. Self-reported dates of vaccination were subject to recall error, with a small proportion reporting their first or last dates of vaccination as 2020. However, a majority (55.8%) remembered their first vaccination as occurring in 2021, which dropped to 36.6% in 2022. Regarding the most recent dose, around half (50.6%) reported in 2021 and 42.9% in 2022, indicating that a significant proportion had not received a vaccine or booster in over a year.

In sub-group analysis, all demographic groups reported higher levels of vaccination than in 2021 (Figure 2). However, disparities persisted from the 2021 survey. Vaccine prevalence was higher among males (69.6%, 95% CI: 64.5–74.8)) vs. females (56.0%, 95% CI: 49.7–62.3); adults aged 50+ years and older (71.2%, 95% CI: 63.1–80.3) vs. younger adults 18–29 (54.4%, 95% CI: 47.2–61.7), and those with university education (74.4%, 95% CI: 68.5–80.4) vs. those with primary or no education (62.6%, 95% CI: 53.8–71.4). Despite this, the gap in vaccination by education status has narrowed compared to that in 2021.

#### 3.2.2. Vaccine Hesitancy

In the overall population, the prevalence of vaccine hesitancy was 31.7% (95% CI: 27.8–35.6) in 2023, a non-significant decrease from 34.1% (95% CI: 28.1–40.6) observed in 2021 (*p* = 0.395) (Appendix A). People residing in refugee camps reported the most favorable attitudes, with 18.5% vaccine-hesitant, followed by rural areas (22.0%), and the highest level of hesitancy was among urban residents (34.2%) (*p* < 0.001).

In the sub-group analysis (Figure 3), lower hesitancy was reported by older adults, males, and people with higher levels of education. Hesitancy was 25.5% (95% CI: 17.1–33.9) among adults aged 50 years old or older compared with 30.7% (95% CI: 23.1–38.3) among those 30–39 years old and was highest (40.1%, 95% CI: 33.0–47.3) among those 18–29 years old. Among men, 25.0% (95% CI: 20.1–30.0) were vaccine-hesitant compared with 39.8% (95% CI: 33.6–46.0) of women. Hesitancy was reported by 21.9% (95% CI: 16.2–27.7) of people with university education compared with 34.4% (95% CI 25.8–43.0) among those with primary or no education and 39.6% (95% CI 33.1–46.1) of those with secondary or vocational school background.

### 3.3. Exposure to Vaccine Promotion Activities

Approximately half of the population was exposed to each of the vaccine promotion activities assessed in the survey. Referral by a health worker was reported by 50.1% (95% CI: 45.8–54.4), and receipt of information by 58.5% (95% CI: 54.3–62.7) of adults. No significant differences were observed in the exposure levels of the strata. In bivariate regression (Table 4 and Table 5), people who were referred reported 5.6 (95% CI: 3.8–8.2) times higher odds of being vaccinated and had 0.2 (95% CI: 0.1–0.3) times the odds of being vaccine-hesitant compared with people who did not report a referral. Among those who received information, the odds of vaccination were 4.2 (95% CI: 2.9–6.1) times higher, and the odds of vaccine hesitancy were 80% lower (OR: 0.2, 95% CI: 0.2–0.4) vs. those who did not receive information.

### 3.4. Trust in the Health System

Overall, trust in the health system with respect to the vaccine was high. More than 80% of people believed that the health system could safely administer the vaccine to the population. Trust was lower among urban residents (80.3%) than among camp-based (92.2%) or rural (92.1%) residents (*p* < 0.001).

When presented with a list of information sources on the COVID-19 vaccine, HCWs were reported as a trusted source of information on the COVID-19 vaccine by the vast majority (76.7%) of people. Similar variations in trust in the health system were observed by strata (*p* = 0.005). Trust in CHWs was half (37%) that of HCWs and was the highest among rural residents (56.8%) (*p* < 0.001).

The odds of being vaccinated were 4.3 times the odds (95% CI: 2.5–7.3) among people who trusted the healthcare system to roll out the vaccine safely, 2.7 times the odds (95% CI: 1.8–4.0) among those who trusted HCWs, and 1.7 times the odds (95% CI: 1.2–2.5) among people who trusted CHWs as a source of vaccine information, compared to individuals who did not trust these sources. The odds of vaccine hesitancy were 80% lower (95% CI: 0.1–0.4) among people who trusted the healthcare system with the rollout, 70% (95% CI: 0.2–0.5) lower among those who trusted HCWs, and 60% (95% CI: 0.3–0.7) lower among people who trusted CHWs, compared to individuals who did not trust these sources.

### 3.5. Vaccine Acceptability

Trends in vaccine acceptability varied by question, with camp-based residents generally reporting higher levels of vaccine acceptability than rural and urban residents. Approximately 73% of the population overall (80.7% of camp-based residents, 77.6% of urban residents, and 71.4% of rural residents) considered the vaccine safe or somewhat safe.

A similar proportion expressed concerns regarding vaccine side effects, with the highest skepticism reported by urban residents. Approximately two-thirds of people believed there were better ways to prevent COVID-19 (63.9%) and believed that it was better to develop natural immunity (68.1%). Among camp-based residents, 40.6% of people believed that there were better ways to prevent COVID-19 versus 68.2% of urban residents and 49.7% of rural residents (*p* < 0.001). Natural immunity was believed to be better than vaccination by 48.5% of camp-based residents versus 71.0% of urban and 67.6% of rural residents (*p* < 0.001).

The odds of being vaccinated were significantly higher among people who considered the vaccine safe or somewhat safe (OR 16.4, 95% CI: 9.6–28.0) and were lower among those who were concerned about side effects (OR 0.3, 95% CI: 0.2–0.5), those who believed there were better ways to prevent COVID-19 (OR 0.2, 95% CI: 0.1–0.2), and those who preferred to develop natural immunity (OR 0.1, 95% CI: 0.08–0.3) compared to those who did not. An opposite trend was observed for vaccine hesitancy. The odds of being vaccine-hesitant were significantly lower among people who considered the vaccine safe or somewhat safe (OR 0.1, 95% CI: 0.04–0.10) and higher among those concerned about side effects (OR 3.1, 95% CI: 1.9–4.9), those who preferred other ways to prevent COVID-19 (OR 7.6, 95% CI: 4.7–12.2), and those who preferred to develop natural immunity (OR 6.5, 95% CI: 3.5–12.2) than those who did not.

### 3.6. Risk Perception

There was a notable gap between people’s perception of being at risk of contracting COVID-19 and their self-perceived risk of experiencing severe disease. While the majority (79.7%, 95% CI: 76.5–82.9) believed they were at risk of contracting COVID-19, only 34.3% (95% CI: 30.1–38.3) believed they were at risk of severe health outcomes. Camp-based residents were the most likely to believe that they were at risk of contracting COVID-19 (88.6%); however, they were the least likely to believe that they were at risk of severe disease. Only 23.7% of camp-based residents, 29.8% of rural residents, and 36.0% of urban residents agreed that they were at risk of serious illness, hospitalization, or death due to COVID-19 (*p* = 0.005).

People who viewed themselves at risk of contracting COVID-19 had 0.64 (95% CI: 0.41–0.98) times the odds of being vaccine-hesitant compared with those who did not consider themselves at risk. People who perceived they were at risk of severe disease if they were to contract COVID-19 had reduced odds of hesitancy (OR 0.75, 95% CI: 0.50–1.11), but this was not statistically significant.

### 3.7. Multivariable Models

Adjusted odds of vaccination were 4.2 (95% CI: 2.6–6.8) times the odds among people referred by health workers to be vaccinated compared to people who were not referred (Table 4). Sex, age, and education were all significantly associated with vaccination status. Perception of the vaccine’s safety retained the strongest association with vaccine status (aOR 16.1, 95% CI: 8.9–29.2). Those who believed they were at risk of severe disease had 1.8 (95% CI: 1.0–3.0) times the odds of being vaccinated. Trust in health workers was associated with 1.9 (95% CI: 1.0–3.4) times higher odds of vaccination, indicating an attenuation of the association after adjusting for the exposure and other risk factors.

Adjusted odds of vaccine hesitancy among people who received information were 0.3 (95% CI: 0.2–0.6) times the odds of those who did not receive information on the vaccine (Table 5). Other factors associated with hesitancy were similar to those associated with vaccination, including lower odds of hesitancy among men and those with a university education. Older age was associated with lower odds of being vaccine-hesitant, but it was not statistically significant, unlike vaccination status. Confidence in the vaccine’s safety had the strongest magnitude of association with hesitancy; people who considered the vaccine safe had 90% lower odds of being hesitant (aOR 0.1, 95% CI: 0.04–0.1). People who believed they were at risk of severe disease were around half as likely to be vaccine-hesitant (aOR 0.6, 95% CI: 0.3–1.0). Trust in health workers was associated with 0.4 (95% CI 0.2–0.7) times the odds of vaccine hesitancy.

## 4. Discussion

Our study highlights COVID-19 vaccine trends in the Gaza Strip and factors correlated with vaccine uptake and hesitancy. Vaccine prevalence increased by approximately 15 percentage points between the surveys in October 2021 (49%) and March 2023 (64%). Our estimate of vaccine coverage is somewhat higher than the Palestine Ministry of Health (MoH) projections from April 2023, where service data showed that 696,514 people had received at least one dose of a COVID-19 vaccine [21]. At that time, this figure represented approximately 57 percent of Gazan adults eligible for the vaccine [13].

Coverage increased in all demographic groups; however, differences in vaccination rates and hesitancy by sex and education persisted. Women and those with less education were still less likely to be vaccinated than men and those with a university education, respectively. The fact that more than a third of adults in Gaza had never received a dose of the COVID-19 vaccine is a concern given the projections from February 2024, which indicated that COVID-19 would be a leading cause of excess mortality due to infectious diseases [1]. Moreover, certain comorbid medical conditions are known risk factors for complications and severe outcomes of COVID-19 disease [22,23]. These conditions include non-communicable diseases, such as hypertension and diabetes, which are highly prevalent in the Gazan population [24].

No decline in vaccine hesitancy was observed between 2021 (34%) and 2023 (32%). The majority (80%) of the non-vaccinated population was classified as hesitant, with an increase from 67% in 2021. Lack of confidence in the vaccine’s safety was the single most important risk factor associated with non-vaccination and vaccine hesitancy in both the 2021 and 2023 surveys. In the multivariable model, belief in the vaccine’s safety was associated with 33 times higher odds of vaccination. The relative importance of safety perception aligns with studies from other settings that found perceived safety to be a central factor in vaccination status [6,25,26].

The saturation of those open to persuasion probably explains the limited growth in vaccine coverage and lack of significant decline in vaccine hesitancy between the 2021 and 2023 surveys. While the early phase of the global vaccine roll-out had been characterized by insufficient supply and notable disparities in coverage, as early as 2022, vaccine campaigns in high-income countries had reached most people who were willing to take the vaccine or who could be convinced with moderate efforts [27,28]. Vaccination rates for COVID-19 and other infectious diseases have stagnated to varying degrees across countries, following a logarithmic growth trajectory, which was also observed in Gaza during this period [27,29]. Moreover, we found a high percentage of people who believed there were better ways to prevent COVID-19 compared to vaccination and who preferred natural immunity, suggesting vaccine mandates could be driving many people’s decisions to become vaccinated.

While our survey design limits the interpretation of differences by strata, we found a higher prevalence of vaccination, a lower prevalence of vaccine hesitancy, and higher levels of trust in HCWs and CHWs among people residing in camps and in rural areas than in urban areas. Although we did not investigate reasons for strata-specific differences, it is plausible that the refugee population’s access to the United Nations Relief Works Agency for Palestinian Refugees in the Near East (UNRWA) health system contributes to higher levels of trust in health workers. It is unclear whether that level of trust will be transferrable to non-UNRWA providers as patients are forced to seek care from other health facilities. Notably, the UNRWA health system had declined from 22 health facilities prior to the October 2023 war to only seven operational facilities in May 2024 [30]. Community-based health services outside of traditional health facilities, such as those in shelters, may need to play a bigger role in healthcare delivery and vaccine services in the near term.

Encouragingly, survey respondents held widely positive views on the health system and health workers. Healthcare workers were the most trusted source of information on the COVID-19 vaccine, as affirmed by 76% of respondents. Both bivariate and multivariable results showed a strong association between people’s exposure to health worker influence (receiving information and/or referral) and better vaccine outcomes. In multivariable models adjusted for demographic factors and risk perceptions, those referred by HCWs for vaccination had 4.1 (95% CI: 2.5–6.7) times the odds of being vaccinated. Those who received information on the vaccine from HCWs had 0.3 (95% CI: 0.2–0.6) times the odds of being hesitant.

These findings are important due to the limited evidence regarding the ability of health workers to influence vaccine hesitancy [31]. Available research suggests that the role of HCWs may be specific to the setting and population. For example, a study in the US found that respondents with a primary care provider were more hesitant towards the COVID-19 vaccine than those without one [32]. In contrast, an online survey experiment found that support for vaccines from medical practitioners, but not religious leaders, reduced COVID-19 vaccine hesitancy [33]. Studies involving other types of vaccines similarly demonstrate variability and uncertainty about HCWs’ influence, which may depend on the level of trust in these professions. A Cochrane review of factors influencing parents’ decisions on childhood immunization found that individual and social factors were key drivers of vaccine acceptance [34]. Interactions with frontline healthcare workers also played a role, but primarily through parents’ perception of positive interpersonal communication rather than the accuracy and quality of the information provided by HCWs [34]. Other systematic reviews found that both parental knowledge about the vaccine and trust in the healthcare profession were associated with childhood vaccination [31,35]. This suggests that improving the provision of sensitive education by HCWs could be influential in certain settings. Our findings in Gaza revealed a positive association between health worker exposures and vaccine outcomes, underscoring the importance of engaging HCWs in promoting vaccines. However, enhancing the quality of providers’ interpersonal communication remains crucial to establishing trust in the profession and, therefore, its influence on vaccination decisions.

The role of CHWs is also relevant in terms of trust and effectiveness of promotion efforts. Studies on lay health workers in vaccine programs suggest CHWs can be moderately effective in promoting vaccines and increasing uptake [36,37]. However, the quality of evidence is mixed, and most research has been conducted in higher-income settings. In Gaza, we found that trust in formal HCWs did not necessarily extend to CHWs; only 37% of community members trusted CHWs to provide accurate vaccine information, compared to 76% for HCWs. Investing in training and skills development of CHWs in Gaza will be crucial to equip them with the knowledge and tools to deliver accurate messages consistent with those provided by HCWs, thereby building trust within their communities.

A key consideration and caveat regarding the role of health professions is that the attitudes and perceptions of HCWs towards vaccines cannot be automatically assumed to be positive. Their influence on vaccine hesitancy may be negative in some cases. In Gaza, our 2021 survey of a non-probability sample of HCWs found that 89.4% were vaccinated, but this occurred in the context of a vaccine mandate. Many HCWs had expressed skepticism about the COVID-19 vaccine. This sentiment is found in global studies of HCWs’ vaccine perceptions and uptake [38]. Factors associated with vaccine hesitancy in HCW populations—such as demographic, occupational, economic, and social influences—mirror those identified in the general population [39]. A study of influenza vaccine hesitancy among HCWs in Jordan found that conspiracy beliefs were linked to lower levels of vaccination, and these beliefs were more commonly held among those in less highly trained professional cadres and among female HCWs [40]. This trend is likely to apply to CHWs, who receive less formal and clinical training. Further exploration of HCWs’ and CHWs’ perceptions of vaccines in Gaza is an important area of future research to inform training programs and vaccination strategies.

This study had several limitations that should be considered when interpreting the results. First, the classification of vaccination status and exposure to vaccine promotion activities relied on self-reported data, which were not independently verified. The observed vaccination prevalence was slightly higher than that reported in MoH administrative records, potentially due to social desirability bias, as respondents were aware that the survey was conducted on behalf of an international health organization.

Second, the cross-sectional survey design restricts the ability to infer causal relationships between health worker interventions and vaccination outcomes. Improved vaccine outcomes cannot be attributed to these exposures because of temporal ambiguities and the absence of a counterfactual condition. Future prospective studies would be valuable in addressing these limitations in other settings, particularly as there is a dearth of intervention research testing vaccine promotion by HCWs in resource-constrained settings.

Finally, comparisons of the 2021 and 2023 results should be made with caution because of differences in the sampling design and the weighting approach. Specifically, the 2023 weighting approach did not account for clustering because survey cluster IDs were not available in the original dataset; full verification of the data was not feasible as the analyses were in progress at the start of the October 2023 war. Nevertheless, a notable strength of our study is the consistent use of the same instrument in both the 2021 and 2023 surveys. The standard questionnaire design enhances the comparability of trends in vaccine outcomes, exposure to interventions, and risk factors within the same population.

## 5. Conclusions

Our study found that approximately one-third of the adult population in Gaza remained unvaccinated in March 2023 despite a modest increase in vaccination prevalence since the previous survey in October 2021. Displaced populations with underlying risk factors are particularly vulnerable to severe COVID-19 infection. Therefore, it is essential that health sector strategies in Gaza include sustained supply and promotion of COVID-19 vaccines. Vaccine strategies should include specific plans to address disparities in vaccine coverage among women, younger adults, those with less education, and people from urban areas.

We observed no improvement in vaccine hesitancy between 2021 and 2023. Lack of trust in the safety of the COVID-19 vaccine continued to drive vaccine refusal and was widespread among unvaccinated individuals. Hesitancy appears to be particularly entrenched in those who have not already been vaccinated. However, our study also suggested that health workers can positively influence vaccine uptake. A large proportion of Gazans reported trusting HCWs for vaccine information, and vaccine promotion efforts by HCWs were associated with improved vaccine outcomes. It is less certain whether CHWs can have the same influence, given the lower level of trust observed compared to trust in HCWs. Investing in the skills development of CHWs will be strategic to bolster the health system’s ability to deliver effective messages on vaccination.

## Figures and Tables

**Figure 1 vaccines-12-01098-f001:**
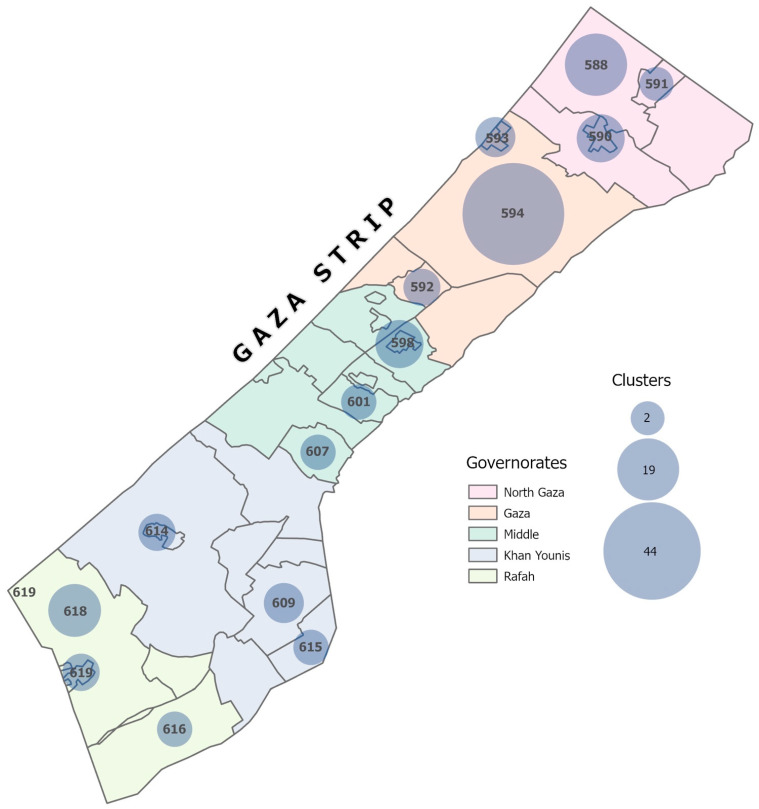
Selected geo-locations for the follow-up survey in Gaza, 2023. Numbered geo-locations on the map represent geo-locations selected from the survey.

**Figure 2 vaccines-12-01098-f002:**
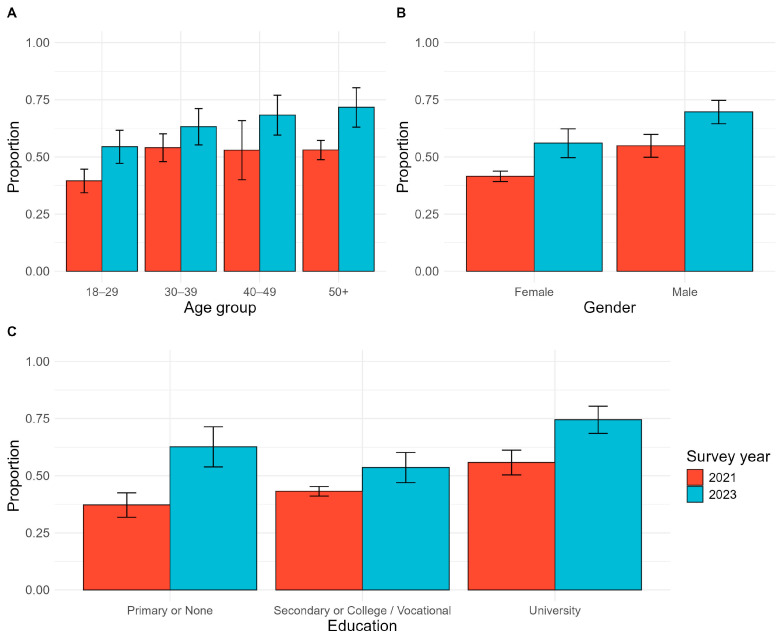
Changes in prevalence of vaccination between 2021 and 2023 by demographic group. (**A**) Vaccination prevalence by age group; (**B**) vaccination prevalence by gender; (**C**) vaccination prevalence by highest level of education.

**Figure 3 vaccines-12-01098-f003:**
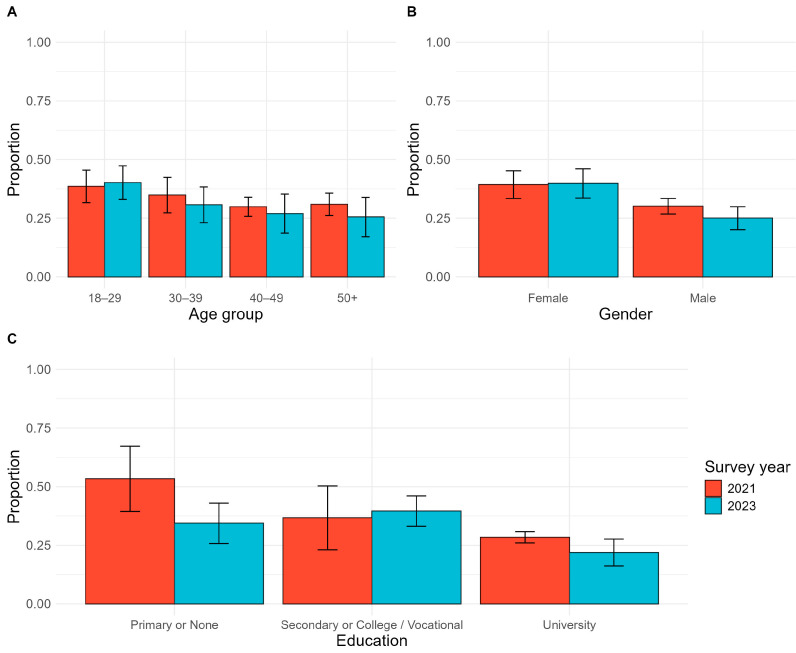
Changes in prevalence of vaccine hesitancy between 2021 and 2023 by demographic group. (**A**) Vaccine hesitancy prevalence by age group; (**B**) vaccine hesitancy prevalence by gender; (**C**) vaccine hesitancy prevalence by highest level of education.

**Table 1 vaccines-12-01098-t001:** Thematic areas investigated in the analysis.

Primary Outcome Measures and Determinants	Variable	Survey Question
Primary Outcomes		
Vaccination status	Vaccinated	“Have you already received the COVID-19 vaccine?” Adult members of the household 18 years or older who received at least one dose of any COVID-19 vaccine prior to the survey were classified as vaccinated.
Vaccine hesitancy	Vaccine hesitant: Lack of intent to receive a COVID-19 vaccine	“If you could get a COVID-19 vaccine this week, would you get it?” Non-vaccinated individuals were classified as hesitant if they responded “no” or “unsure”, while all vaccinated individuals were classified as non-hesitant.
Exposure to vaccine promotion activities		
Referral	Referred by a healthcare provider to take the vaccine	“Did IMC partner healthcare organizations or CBOs refer you to take the vaccine?”
Information	Received information from a healthcare provider about the vaccine	“Did you receive any information about the vaccine from IMC partner healthcare organizations or CBOs?”
Determinants		
Demographic variables	Sex Age Governorate Highest level of education	Respondent gender (female/male/other/prefer not to say) Respondent age (18–29, 30–39, 40–49, 50+) Governorate (North, Gaza, Middle, Khan Younis, Rafah) “What is the highest educational grade/level that you have completed?” (none, primary, secondary, vocational, university)
Trust in the health system	Believe the health system can safely administer the vaccine to the population Health workers are a trusted source of information on the vaccine Community health workers are a trusted source of information on the vaccine	“Do you think your health system can safely administer the COVID-19 vaccine to the population?” “Which source do you trust?” [to provide information about the COVID-19 vaccine] Respondents were asked to report yes/no for each of the following: HCWs at health clinics; CHWs; community leaders, radio, television, newspapers, mass events, and local leaders.
Vaccine acceptability	Considers vaccine safe or somewhat safe Concerned about the risk of side effects Believe there are better ways to prevent COVID-19 than vaccination Think it is better to get COVID-19 and develop natural immunity than to get the vaccine	“Do you consider the COVID-19 vaccines safe?” “Are you concerned about any risks or side effects with the COVID-19 vaccine?” “Do you believe that there are other (better) ways to prevent COVID-19 instead of the vaccine?” “Do you think it is better to get COVID-19 and develop natural immunity than to get the vaccine?”
Perceived risk	Self-perceived risk to get COVID-19 infection Self-perceived risk to develop severe disease following COVID-19 infection	“Do you think you are at risk to get COVID-19?” “Do you think you can get seriously ill, hospitalized or die if you get COVID-19?”

HCWs = healthcare workers, CHWs = community health workers, CBO = community-based organizations.

**Table 2 vaccines-12-01098-t002:** Demographic characteristics of COVID-19 survey respondents in Gaza, 2023.

Variable	Frequency (N = 894)	% ^1^ (95% CI) ^2^
**Age**		
18–29	279	31.7 (27.8–35.6)
30–39	230	24.4 (20.8–28.0)
40–49	185	19.8 (16.3–23.3)
50+	200	24.2 (20.3–28.0)
**Sex**		
Female	365	45.0 (40.8–49.1)
Male	528	55.0 (50.9–59.2)
**Highest Education**		
Primary or None	190	18.9 (15.7–22.2)
Secondary or College/Vocational	347	41.9 (37.6–46.1)
University	357	39.2 (35.0–43.4)
**Governorate**		
*Gaza*	317	34.2 (32.1–36.3)
*Khan Younis*	172	19.8 (16.8–22.8)
*Middle*	125	13.6 (10.4–16.8)
*North*	176	20.0 (18.4–21.5)
*Rafah*	104	12.5 (11.7–13.2)
**Strata**		
Camp	309	12.6 (10.9–14.3)
Rural	104	4.5 (3.6–5.4)
Urban	481	82.9 (80.9–84.9)

^1^ Weighted percentage. ^2^ CI = Confidence Interval.

**Table 3 vaccines-12-01098-t003:** Stratified, weighted prevalence estimates of primary outcomes, exposures, and risk factors.

Characteristic	Overall, N = 894		Camp, N = 309		Rural, N = 104		Urban, N = 481		*p*-Value ^3^
	n ^1^	% (95% CI) ^2^	n ^1^	% (95% CI) ^2^	n ^1^	% (95% CI) ^2^	n ^1^	% (95% CI) ^2^	
**Vaccine outcomes**									
Received Vaccine	597	63.5 (59.4–67.5)	236	73.9 (68.1–79.7)	71	71.2 (62.3–80.2)	290	61.5 (56.7–66.3)	0.001
Vaccine Hesitant	248	31.7 (27.8–35.6)	56	18.5 (13.5–23.4)	26	22.0 (13.8–30.1)	166	34.2 (29.5–38.9)	<0.001
**Exposure to vaccine promotion by health workers**									
Referred to get vaccine	430	50.1 (45.8–54.4)	147	43.8 (38.1–49.4)	53	51.3 (41.1–61.6)	230	51.0 (45.9–56.1)	0.2
Received information on vaccine	514	58.5 (54.3–62.7)	184	57.0 (50.8–63.1)	61	62.4 (52.4–72.4)	269	58.5 (53.6–63.5)	0.7
**Trust in health system**									
Believe the health system can safely administer the vaccine to the population	770	82.3 (78.9–85.7)	289	92.2 (88.2–96.2)	98	92.1 (85.7–98.4)	383	80.3 (76.3–84.3)	<0.001
Healthcare workers are a trusted source of information on the vaccine	683	76.7 (73.3–80.0)	250	82.3 (77.5–87.1)	90	87.8 (81.2–94.4)	343	75.2 (71.3–79.2)	0.005
Community health workers are a trusted source of information on the vaccine	338	37.1 (33.2–40.9)	109	42.4 (36.5–48.2)	51	56.8 (47.0–66.6)	178	35.2 (30.6–39.8)	<0.001
**Vaccine acceptability**									
Considers vaccine safe or somewhat safe	673	72.9 (69.1–76.6)	253	80.7 (75.6–85.8)	80	77.6 (69.2–86.0)	340	71.4 (67.0–75.9)	0.015
Concerned about risk of side effects	627	70.4 (66.6–74.1)	227	72.9 (67.3–78.4)	79	75.3 (66.3–84.3)	321	69.7 (65.3–74.2)	0.4
Believe there are better ways to prevent COVID-19 than vaccination	500	63.9 (59.9–67.9)	119	40.6 (34.6–46.6)	58	49.7 (39.5–59.8)	323	68.2 (63.6–72.8)	<0.001
Think it is better to get COVID-19 and develop natural immunity than to get the vaccine	551	68.1 (64.0–72.1)	129	48.5 (42.4–54.6)	73	67.6 (57.9–77.2)	349	71.0 (66.3–75.8)	<0.001
**Risk perception**									
Think you are at risk to get COVID-19	711	76–83	266	88.6 (85.1–92.2)	80	84.3 (78.3–90.3)	365	78.1 (74.3–81.9)	<0.001
Think you can get seriously ill, hospitalized, or die if you get COVID-19	253	30–38	55	23.7 (17.7–29.8)	27	29.8 (20.1–39.5)	171	36.0 (31.2–40.8)	0.005

^1^ Frequency. ^2^ Weighted %, CI = Confidence Interval. ^3^ chi-squared test with Rao and Scott’s second-order correction.

**Table 4 vaccines-12-01098-t004:** Unadjusted and adjusted odds ratios for vaccination status.

Characteristic	Unadjusted (N = 894)			Adjusted (N = 876)		
	OR ^1^	95% CI	*p*-Value	aOR ^2^	95% CI	*p*-Value
**Exposure to vaccine promotion by health workers**						
Referred to get vaccine	5.58	3.79, 8.21	<0.001	4.20	2.59, 6.83	<0.001
Received information on vaccine	4.20	2.88, 6.12	<0.001	—	—	—
**Demographics**						
Sex			0.001			
Female	—	—		—	—	
Male	1.80	1.27, 2.57		1.80	1.12, 2.91	0.016
Age			0.006			
18–39	—	—		—	—	
40+	1.68	1.16, 2.43		1.87	1.10, 3.18	0.021
Highest education			<0.001			
Primary or None	—	—		—	—	
Secondary or College/Vocational	0.69	0.43, 1.09		0.78	0.39, 1.53	0.46
University	1.74	1.07, 2.84		2.81	1.42, 5.54	0.003
**Trust in health system**						
Believe the health system can safely administer the vaccine to the population	4.25	2.49, 7.27	<0.001			
Healthcare workers are a trusted source of information on the vaccine	2.69	1.83, 3.96	<0.001	1.85	1.02, 3.39	0.044
Community health workers are a trusted source of information on the vaccine	1.71	1.18, 2.47	0.004			
**Vaccine acceptability**						
Considers vaccine safe or somewhat safe	16.4	9.55, 28.0	<0.001	16.1	8.88, 29.2	<0.001
Concerned about risk of side effects	0.32	0.21, 0.49	<0.001			
Believe there are better ways to prevent COVID-19 than vaccination	0.16	0.10, 0.24	<0.001			
Think it is better to get COVID-19 and develop natural immunity than to get the vaccine	0.14	0.08, 0.25	<0.001			
**Risk perception**						
Think you are at risk to get COVID-19	1.36	0.89, 2.06	0.15			
Think you can get seriously ill, hospitalized, or die if you get COVID-19	1.38	0.94, 2.02	0.10	1.75	1.01, 3.02	0.045

^1^ OR = Odds Ratio, CI = Confidence Interval. ^2^ aOR = Adjusted Odds Ratio, CI = Confidence Interval.

**Table 5 vaccines-12-01098-t005:** Unadjusted and adjusted odds ratios for vaccine hesitancy.

Characteristic	Unadjusted (N = 894)			Adjusted (N = 873)		
	OR ^1^	95% CI	*p*-Value	aOR ^2^	95% CI	*p*-Value
**Exposure to vaccine promotion by health workers**						
Referred to get vaccine	0.17	0.11, 0.25	<0.001	—	—	—
Received information on vaccine	0.23	0.16, 0.35	<0.001	0.34	0.20, 0.57	<0.001
**Demographics**						
**Sex**			<0.001			
Female	—	—		—	—	
Male	0.50	0.35, 0.73		0.50	0.30, 0.83	0.007
**Age**						
18–39	—	—		—	—	
40+	0.63	0.43, 0.92		0.63	0.34, 1.15	0.13
**Highest education**			<0.001			
Primary or None	—	—		—	—	
Secondary or College/Vocational	1.25	0.78, 2.01		1.26	0.60, 2.65	0.53
University	0.54	0.32, 0.89		0.41	0.20, 0.87	0.021
**Trust in health system**						
Believe the health system can safely administer the vaccine to the population	0.22	0.13, 0.38	<0.001			
Healthcare workers are a trusted source of information on the vaccine	0.33	0.22, 0.48	<0.001	0.38	0.20, 0.72	0.003
Community health workers are a trusted source of information on the vaccine	0.44	0.29, 0.66	<0.001			
**Vaccine acceptability**						
Considers vaccine safe or somewhat safe	0.06	0.04, 0.10	<0.001	0.07	0.04, 0.12	<0.001
Concerned about risk of side effects	3.08	1.94, 4.91	<0.001			
Believe there are better ways to prevent COVID-19 than vaccination	7.56	4.69, 12.2	<0.001			
Think it is better to get COVID-19 and develop natural immunity than to get the vaccine	6.49	3.47, 12.2	<0.001			
**Risk perception**						
Think you are at risk to get COVID-19	0.64	0.41, 0.98	0.039			
Think you can get seriously ill, hospitalized, or die if you get COVID-19	0.75	0.50, 1.11	0.14	0.55	0.30, 1.00	0.049

^1^ OR = Odds Ratio, CI = Confidence Interval. ^2^ aOR = Adjusted Odds Ratio, CI = Confidence Interval.

## Data Availability

The datasets used and analyzed during the current study are available from the corresponding author’s institution upon reasonable request.

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
