# Peer review of "COVID-19 Vaccination and Vaccine Hesitancy in the Gaza Strip from a Cross-Sectional Survey in 2023: Prevalence, Risk Factors, and Associations with Health System Interventions"

_vaccines, 2024, doi:10.3390/vaccines12101098_

Round 1

Reviewer 1 Report

Comments and Suggestions for Authors

This is a very interesting study on activities to promote vaccination. Although the research is based on a questionnaire, the content of the survey and the method of investigation are well-conceived. The results are an interesting reminder of the difficulties involved in promoting vaccination, but they are also interesting in that they reaffirm the importance of health care workers. In the discussion, other similar studies are taken up and compared (line 366-378). However, the only comparison target is the COVID-19 vaccine, and if you add comparisons and considerations with similar studies related to other vaccination promotions, it will be easier to understand and the issues will become clearer. If possible, I would like you to consider the similarities and differences between the issues related to other vaccination promotions and this COVID19 vaccination promotion.

Although authors mentioned healthcare workers (HCWs) and community healthcare workers (CHWs) in the main text, it is difficult to understand how they are used.

If there is any importance in making a distinction, it seems that some explanation is necessary.

Author Response

Reviewer: 1 

This is a very interesting study on activities to promote vaccination. Although the research is based on a questionnaire, the content of the survey and the method of investigation are well-conceived. The results are an interesting reminder of the difficulties involved in promoting vaccination, but they are also interesting in that they reaffirm the importance of health care workers.

In the discussion, other similar studies are taken up and compared (line 366-378). However, the only comparison target is the COVID-19 vaccine, and if you add comparisons and considerations with similar studies related to other vaccination promotions, it will be easier to understand and the issues will become clearer. If possible, I would like you to consider the similarities and differences between the issues related to other vaccination promotions and this COVID₋19 vaccination promotion.

Authors’ Response: Thank you for this feedback, it has helped us to strengthen the manuscript with comparisons to non-COVID-19 vaccines. We have added to the discussion section a comparison of health workers’ role in parental decisions on vaccination, as below:

Studies involving other types of vaccines similarly demonstrate variability and uncertainty about HCWs’ influence, which may depend on the level of trust in these professions. A Cochrane review of factors influencing parents’ decisions on childhood immunization found that individual and social factors were key drivers of vaccine acceptance [34]. Interactions with frontline healthcare workers also played a role, but primarily through parents’ perception of positive interpersonal communication, rather than the accuracy and quality of the information provided by HCWs [34]. Other systematic reviews found that both parental knowledge about the vaccine and trust in the healthcare profession was associated with childhood vaccination [31, 35]. This suggests that improving provision of sensitive education by HCWs could be influential in certain settings.

We have also elaborated on health workers’ perceptions of vaccines, with reference to other types of vaccines.

Factors associated with vaccine hesitancy in HCW populations – such as demographic, occupational, economic, and social influences – mirror those identified in the general population [39]. A study of influenza vaccine hesitancy among HCWs in Jordan found that conspiracy beliefs were linked to lower levels of vaccination, and these beliefs were more commonly held among those in less highly trained professional cadres and among female HCWs [40].

Although authors mentioned healthcare workers (HCWs) and community healthcare workers (CHWs) in the main text, it is difficult to understand how they are used. If there is any importance in making a distinction, it seems that some explanation is necessary.

Authors’ Response: Thank you for this feedback. We have added to the introduction a clarification of the terms along with references.

In our study, we distinguish between formal health care workers (HCWs) and community health workers (CHWs). HCWs are formally trained providers, such as physicians, nurses and midwives. In contrast, CHWs are community members who, after receiving basic non-clinical training, work either voluntarily or for a stipend to support public health messaging and case identification [8, 9].

Of note, CHWs were recruited under the COVID-19 response project referenced in the study, in order to undertake the following:

  1. Screening for COVID-19 at entrances of local partner organizations and health facilities
  2. Registering visitors and referring suspected cases for testing 
  3. Community awareness raising on COVID-19 prevention in the facility and within the community 

Reviewer 2 Report

Comments and Suggestions for Authors

The stuyd is well-presented and it has significant findings. However, I have some minor suggestions.

1. Please add the vaccine status of the included subjects in more detail, such as unvaccinated, full vaccinated, booster vaccinated.

2. Please add the history of COVID-19 of the study subjects, such as primary infeciton or re-infection.

Author Response

The stuyd is well-presented and it has significant findings. However, I have some minor suggestions.

  1. Please add the vaccine status of the included subjects in more detail, such as unvaccinated, full vaccinated, booster vaccinated.

Authors’ Response: Thank you for this feedback. Additional details on respondents’ vaccination history have been added to the results section and as supplementary files 3 and 4.

The majority of vaccines administered were Pfizer (63.6%) and Sputnik (24.7%), either alone or in combination. To be classified as fully vaccinated, at least two doses were required. A sizable proportion were only partially vaccinated, as 43.5% of respondents reported receiving one dose, 48.6% reported receiving two doses, and 7.9% reported receiving three doses. Self-reported dates of vaccination were subject to recall error, with a small proportion reporting their first or last dates of vaccination as 2020. However, a majority (55.8%) remembered their first vaccination as occurring in 2021, which dropped to 36.6% in 2022. Regarding the most recent dose, around half (50.6%) reported 2021 and 42.9% in 2022, indicating that a significant proportion had not received a vaccine or booster in over a year.

  1. Please add the history of COVID-19 of the study subjects, such as primary infeciton or re-infection.

Authors’ Response:

Regarding comment 2, we did not collect data on respondents’ COVID-19 history based on contextual understanding that cases were under-detected in Gaza (as in many countries) due to limited uptake of testing for those with mild cases.

Reviewer 3 Report

Comments and Suggestions for Authors

Authors describe a demographic survey of several populations of individuals in the Gaza strip on various questions related to COVID 19 vaccination.  The results are similar to previous reports in other areas that have conducted similar surveys.  This survey is unique as it covers various issues related to COVID 19 vaccination in 2021 and then again in 2023 during the wars occurring in these areas.

Data and results are not surprising as there seems to be a correlation between education, gender and sites of residence and vaccine hesitancy, confidence in the vaccine, influence of health care workers on whether individuals receive the vaccine and whether individuals are living in rural, urban and in this study refugee camps.  The data from the refugee camps is interesting and presents new evidence in the literature.

Overall, a good paper with information that will be of interest to individuals in the middle east and especially in Gaza.

Author Response

Authors describe a demographic survey of several populations of individuals in the Gaza strip on various questions related to COVID 19 vaccination.  The results are similar to previous reports in other areas that have conducted similar surveys.  This survey is unique as it covers various issues related to COVID 19 vaccination in 2021 and then again in 2023 during the wars occurring in these areas.

Data and results are not surprising as there seems to be a correlation between education, gender and sites of residence and vaccine hesitancy, confidence in the vaccine, influence of health care workers on whether individuals receive the vaccine and whether individuals are living in rural, urban and in this study refugee camps.  The data from the refugee camps is interesting and presents new evidence in the literature.

Overall, a good paper with information that will be of interest to individuals in the middle east and especially in Gaza.

Authors’ Response: Thank you for this feedback.

Reviewer 4 Report

Comments and Suggestions for Authors

The manuscript by Majer and colleagues investigates COVID-19 vaccination and vaccine hesitancy among residents of the Gaza Strip in 2023. The methods are well-designed and the authors discuss the limitations of the study.  Overall, this study is well-written with the main conclusion being that 63.5% of adults were vaccinated in 2023 versus 49.1 % in 2021. However, the level of vaccine hesitancy is nearly the same (34.1% in 2021 versus 34.1% in 2023).  How these findings will lead to better vaccination strategies is discussed. 

Comments on the Quality of English Language

The English is fine.

Author Response

The manuscript by Majer and colleagues investigates COVID-19 vaccination and vaccine hesitancy among residents of the Gaza Strip in 2023. The methods are well-designed and the authors discuss the limitations of the study.  Overall, this study is well-written with the main conclusion being that 63.5% of adults were vaccinated in 2023 versus 49.1 % in 2021. However, the level of vaccine hesitancy is nearly the same (34.1% in 2021 versus 34.1% in 2023).  How these findings will lead to better vaccination strategies is discussed. 

Authors’ Response: Thank you for this feedback.

Round 2

Reviewer 1 Report

Comments and Suggestions for Authors

Authors responded well to my questions and revised manuscript is considered to be worthy of acceptance.